# Exact Model Order Reduction for the Full-System Finite Element Solution of Thermal Elastohydrodynamic Lubrication Problems

Jad Mounayer 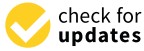 and Wassim Habchi *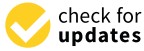

Department of Industrial and Mechanical Engineering, Lebanese American University, Byblos 36, Lebanon
* Correspondence: wassim.habchi@lau.edu.lb

**Abstract:** The derivation of fast, reliable, and accurate modeling procedures for the solution of thermal elastohydrodynamic lubrication problems is a topic of significant interest in the Tribology community. In this paper, a novel model order reduction technique is introduced for the analysis of thermal elastohydrodynamic lubrication problems. The method uses static condensation to reduce the size of the linear elasticity part within the overall matrix system, followed by a splitting algorithm to avoid the burden of solving a semi-dense matrix system. The results reveal the exactness of the proposed methodology, which does not introduce any additional model-reduction approximations to the overall solution. They also reveal the reduction in computational times, which is in the order of 10–20% for line contacts, while it is in excess of 50% for circular contacts. The robustness of the proposed method is displayed by using it to model some relatively highly loaded contacts whose numerical solution is known to be rather challenging.

**Keywords:** thermal elastohydrodynamic lubrication; finite elements; model order reduction; static condensation

## 1. Introduction

Elastohydrodynamic lubrication (EHL) is a full-film lubrication regime where two mechanical components are separated using a high-viscosity fluid—known as the lubricant—which is exposed to sufficiently high pressures to induce elastic deformation of the solid elements. It is usually found in spur gears, roller-element bearings, etc. The significance of this regime mainly lies in reducing the energy consumption of the machine by abating frictional dissipation. In addition, it reduces the risk of damage by preventing metal-to-metal contact between the machine's components.

At the contact level, both mechanical components can be approximated by ellipsoids, and the contact is generally assumed to be a "point contact" where the elastic deformation, pressure, and film thickness can vary in both directions of the contact area. However, according to the application, the contact can be assimilated into a "line contact". In spur gears or roller-element bearings, for example, one of the directions can be considered infinitely long, which will make any gradient in this direction negligible, reducing the dimension of the problem by one.

Additional assumptions are usually adopted in modelling EHL problems. Under certain conditions (low sliding speeds and/or low loads), EHL is considered an isothermal Newtonian process, therefore assuming a Newtonian lubricant response and negligible temperature gradients. Thus, lubricant viscosity is assumed to be constant across the lubricant film thickness. This approximation was long used and appeared to be valid, especially for predicting lubricant film thickness and pressure. However, this conjecture faces some difficulties when it comes to predicting film thickness for either high velocities and/or high loads because of the manifestation of non-Newtonian and thermal effects. These would lead to lubricant viscosity and density variations across the film thickness. It also fails spectacularly in predicting friction under these conditions.

Taking density and viscosity variations across the lubricant film thickness into consideration would shift the regime to a thermal elastohydrodynamic (TEHL) one, which is much more accurate in predicting the lubricant film thickness, as well as frictional dissipation, especially for the different cases mentioned above. The effect of temperature variations in EHL line contacts was first studied theoretically by Cheng [1,2]. A numerical model for point contacts was developed and solved by Zhu and Wen [3] after nearly two decades while assuming a Newtonian Response. This solution was succeeded by the work of Guo et al. [4], Liu et al. [5], and Kim and Sadeghi [6], who ascertained the temperature gradients for Newtonian and non-Newtonian lubricant behavior.

It cannot be denied that the studies mentioned above significantly improved the predictive accuracy of EHL solutions, especially compared to traditional EHL models (i.e., Isothermal Newtonian). However, the main drawback of these methods would be increasing the size of their matrix system since the number of degrees of freedom (dofs) associated with the temperature domain is considerably high. A simplification consisting of reducing the dimension of the fluid temperature domain by one by assuming a parabolic temperature profile across the lubricant film thickness was first adopted for line contacts by Salehizadeh and Saka [7], Wolff and Kubo [8], and also Kazama et al. [9]. The same simplification was applied to point contacts by Kim et al. [10,11], Jiang et al. [12], and also Lee et al. [13]. However, Kazama et al. [9] showed that this assumption led to a significant decrease in the solution accuracy, especially for temperature variations at the contact inlet, because of the occurrence of complex inlet reverse flows. Moreover, the TEHL analysis involves an additional complexity that is related to the evaluation of cross-film integral terms to account for density and viscosity variations. The supplementary size and complexity of the matrix system will both contribute to a remarkable increase in the associated computational overhead. This becomes even worse when a weak coupling technique is used. In fact, several early papers such as the pioneering work of Dowson and Higginson [14] and the more developed work of Hamrock and Dowson [15] revealed different difficulties faced when solving EHL problems. Solving the equations using a weak-coupling technique (i.e., non-synchronized resolution of the different inherent equations) usually leads to slow convergence rates.

To diminish the computational overhead associated with weak coupling techniques, several works tried to develop a full-coupling technique, where all equations are solved simultaneously, therefore leading to faster convergence rates. Rohde and Oh [16] were among the first researchers to tackle this problem. Even though their developed work in [17] converged rapidly with only a few iterations, their method suffered from a main drawback, which is the density of the arising matrix system. In fact, the solid elastic deformation field was resolved based on the half-space theory, which evaluates the deformation using integral terms that relate each point of the discretized computational domain to all other points, resulting in a dense Jacobian matrix. In addition, their method was only valid for light and moderate loads. Furthermore, the simultaneous update of all pressures at all discretization points meant tedious treatment of the free cavitation boundary arising at the outlet of the contact. This encouraged Holmes et al. [18] to introduce a new model to evaluate elastic deformation based on the finite element method (FEM). This model featured sparse matrices, derived using the half-space theory to evaluate the elastic deformations. However, especially for point contacts, the matrix system still had a considerable bandwidth requiring a special technique to be solved. On the other hand, other works such as Bruyere et al. [19] used Computational Fluid Dynamics (CFD) to solve the Navier–Stokes equations instead of the simplified Reynolds equations for the fluid part, and linear elasticity equations for the solid part using a full-system approach, which was also associated with prohibitively high computational overhead.

Recently, Habchi et al. [20–22] introduced an approach that solves the constraints mentioned above. First of all, the authors proposed solving the problem using a fully coupled scheme, which assured a high rate of convergence. It was also proposed to use classical linear elasticity theory for modeling the elastic domains, where each discretization

node is only related to the other nodes in the same element, which leads to a relatively sparse Jacobian matrix. In addition, stabilizing terms were added, extending the method for high loads up to several Gigapascals. The cavitation boundary condition was treated with a straightforward penalty technique developed by Wu [23].

Despite the simplicity of the model mentioned above, in terms of computational overhead, the model implied the extension of the computational domain into the depth of the solids. Even though the proposed model was optimized using a non-regular non-structured meshing to achieve comparable performance to state-of-the-art existing models, a major improvement was still available since the matrices are still considerably large, especially for thermal point contact cases. In fact, deformations at nodes within the depth of the solid domain are computed in vain since the deformation is only needed on the contact surface. This encouraged the development of several techniques—known as Model Order Reduction (MOR)—to decrease the size of the matrices.

Firstly, the "EHL-basis technique" was proposed by Habchi et al. [24] to reduce the elastic domain. This technique was improved by Maier et al. [25,26] who reduced the hydrodynamic domain to obtain even faster results. A novel reduction technique was also introduced by Scurria et al. [27] based on Galerkin Projections for the structural part and a hyper-reduction for the Reynolds equation. Even though these MOR techniques are very efficient when it comes to computational speed, they face some significant downsides. Primarily, these techniques cannot be generalized. The procedure consists of an exhausting "Offline" phase, which is designed for a specific configuration. The papers cited previously, for example, assumed isothermal Newtonian conditions. Consequently, if any new features are to be considered, such as thermal effects, surface roughness, and non-Newtonian effects, the reduced solution space should be redefined to account for the new features. Additionally, the definition of a new solution space necessitates a high level of expertise, which may make its use even more complicated for novice users. Lastly, the newly defined reduced solution space is constructed from linear combinations of full model solutions (i.e., mode superposition), which are, in turn, an approximation of the exact solution. In addition to the additionally introduced approximations, mode superposition results in micro-oscillations in the obtained solution. This can be considered a minor complication though since deviations between the reduced and full solutions can be negligible, provided a careful selection of the basis functions of the reduced solution space.

Recently, Habchi et al. [28] introduced the Static Condensation with Splitting (SCS) technique to solve the isothermal steady-state EHL problem. It was mainly divided into two parts: First, a "static condensation" or "Guyan condensation" [29], which is used to reduce the size of the matrices by eliminating all the unnecessary nodes in the elastic domain and injecting their effect into the needed ones. The only drawback of this method is that the matrix obtained is dense compared to the sparse matrices usually obtained in finite elements. Consequently, a "Splitting" algorithm was introduced to maintain a standard finite-element sparsity pattern. Even though this method leads to a lesser reduction, it mitigates the limitations mentioned above since a complete solution space is used, which retains its generality. In fact, injecting the effect of the eliminated nodes in the needed ones keeps the solution exact compared to the full model, i.e., no additional MOR-inherent approximations. Moreover, this method is rather simple, only requiring some basic knowledge of linear algebra.

Following the extension of this method to the transient isothermal Newtonian line contact EHL problem with Habchi [30], the current paper details how the computational overhead for the TEHL problem can be substantially reduced by using the SCS technique while taking into consideration the temperature variations across the film thickness and in the solids. This method will be applied for both line and circular contacts since the matrices in both cases have large sizes when temperature gradients are considered. In addition, the fully coupled scheme introduced by Habchi [31] for TEHL problems will be used for enhanced convergence rates.

## 2. Governing Equations

The following section describes the governing equations of the TEHL problem. Assuming a fully-flooded regime where the surfaces are completely separated by the lubricant, both line and circular contacts are considered. A line contact occurs when the two components have an infinite radius of curvature in one spatial direction. Thus, the geometry can be reduced to that of an elastic cylinder of radius *R* in contact with a rigid flat plane as shown in Figure 1 (left). On the other hand, point contact occurs when both components are ellipsoidal. For equal radii of curvature in the *x*- and *y*- directions (i.e., spherical solids), the contact is circular, and its geometry can be reduced to that of an elastic ball of radius *R* in contact with a rigid flat plane as shown in Figure 1 (right). Throughout this paper, subscripts *1* and *2* denote the flat plane and cylinder/ball, respectively, while subscript *f* denotes the fluid/lubricant. Out of simplicity, only circular contacts are considered here with unidirectional surface velocities $u_1$ and $u_2$ in the *x*-direction ($v_1 = v_2 = 0$). However, the proposed methodology may be extended to cover elliptical contacts in a straightforward manner. It is also important to note that only steady-state operation is studied by considering constant velocities $u_1$ and $u_2$ in the x-direction, as well as a constant applied load *F*. All equations are provided using dimensionless variables defined using Hertzian dry contact parameters as a function of the elastic material properties $(E_1, v_1)$ and $(E_2, v_2)$ of the two contacting solids as follows:

$$\text{Line Contacts}: \quad a = \sqrt{\frac{4RF}{\pi E}} \quad and \quad p_h = \frac{2F}{\pi a}$$
$$\text{Circular Contacts}: a = \sqrt[3]{\frac{3RF}{4E}} \quad and \quad p_h = \frac{3F}{2\pi a^2} \tag{1}$$
$$\text{Where}: E = \frac{1}{\frac{1-v_1^2}{E_1} + \frac{1-v_2^2}{E_2}}$$

where *a* is the Hertzian contact half-width or radius for line and circular contacts, respectively, and $p_h$ is the maximum Hertzian pressure. The contact dimensionless operating parameters are defined as follows:

$$X = \frac{x}{a}, \quad Y = \frac{y}{a}, \quad Z = \frac{z}{a} \text{ (Solids) or } Z = \frac{z}{h} \text{ (Lubricant Film)}$$
$$H = \frac{hR}{a^2}, \quad U = \frac{uR}{a^2}, \quad V = \frac{vR}{a^2}, \quad W = \frac{wR}{a^2}, P = \frac{p}{p_h} \tag{2}$$
$$\overline{T} = \frac{T}{T_0}, \quad \overline{\rho} = \frac{\rho}{\rho_R}, \quad \overline{\eta} = \frac{\eta}{\eta_R}, \overline{\tau}_{zx} = \frac{\tau_{zx}}{\tau_0}, \overline{\tau}_{zy} = \frac{\tau_{zy}}{\tau_0}$$

where $\rho_R$ and $\eta_R$ are the lubricant density and viscosity, respectively, calculated at the reference temperature, pressure, and shear stress, *h* is the lubricant film thickness, *u*, *v*, and *w* are the *x*-, *y*-, and *z*-components of the solid elastic deformation field, *p* and *T* are the lubricant pressure and temperature, and $\tau_{zx}$ and $\tau_{zy}$ are the *x*- and *y*-components of the lubricant shear stress. Moreover, $T_0$ corresponds to the ambient temperature.

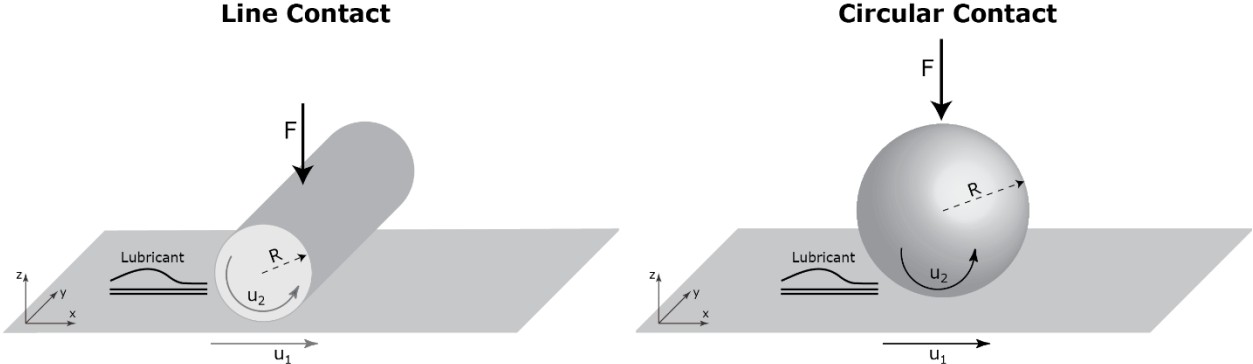

**Figure 1.** Geometry of a line contact (**left**) and circular contact (**right**).

### 2.1. Line Contact

The governing equations for the line contact are detailed in this section. These equations mainly consist of the linear elasticity, hydrodynamic, and load balance equations followed by the conservation of energy equations to study temperature variations, and the shear stress equation that governs shear stress variations throughout the lubricating film.

Since, for line contacts, the conjunction is considered to have an infinite length in the $y$-direction, gradients in this direction are considered negligible. Therefore, the solid computational domain is a square, whereas the contact zone $\Omega_c$ is located on the upper side. It is important to note here that the dimensionless side length of the square was carefully chosen to be 60 according to [21] who proved that this length is sufficient to attain a half-space configuration. The dimensionless length of the contact zone, on the other hand, was chosen to be 6 as shown in Figure 2 going from $X = -4.5$ (inlet) to $X = 1.5$ (outlet).

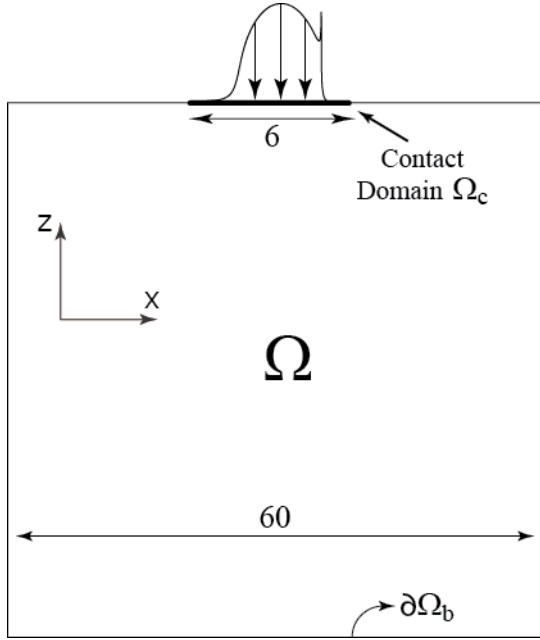

**Figure 2.** Computational domain of a line contact (EHL part).

The first governing equation to be stated is the generalized Reynolds equation [32] applied to the 1D contact domain $\Omega_c$, as introduced by Yang and Wen [33]. It governs the dimensionless hydrodynamic pressure $P$:

$$-\frac{\partial}{\partial X}\left(\bar{\varepsilon}\,\frac{\partial P}{\partial X}\right) + \frac{\partial(\bar{\rho}^*H)}{\partial X} = 0$$

Where :

$$\bar{\varepsilon} = \left(\frac{\bar{\rho}}{\bar{\eta}}\right)_e \frac{H^3}{\lambda}, \quad \left(\frac{\bar{\rho}}{\bar{\eta}}\right)_e = \frac{\bar{\eta}_e\bar{\rho}'_e}{\bar{\eta}'_e} - \bar{\rho}''_e \qquad \text{and } \lambda = \frac{u_m R^2 \eta_R}{a^3 p_h} \text{ with } u_m = \frac{u_1+u_2}{2}$$

$$\bar{\rho}^* = \frac{\bar{\rho}'_e\bar{\eta}_e(u_2-u_1)+\bar{\rho}_e u_1}{u_m}, \qquad\qquad \bar{\rho}_e = \int_0^1 \bar{\rho}\,dZ \tag{3}$$

$$\bar{\rho}'_e = \int_0^1 \bar{\rho} \int_0^Z \frac{dZ'}{\bar{\eta}}\,dZ, \qquad\qquad \bar{\rho}''_e = \int_0^1 \bar{\rho} \int_0^Z \frac{Z'dZ'}{\bar{\eta}}\,dZ$$

$$\frac{1}{\bar{\eta}_e} = \int_0^1 \frac{dZ}{\bar{\eta}}, \qquad\qquad \frac{1}{\bar{\eta}'_e} = \int_0^1 \frac{Z\,dZ}{\bar{\eta}}$$

where $Z = 0$ and $Z = 1$ correspond to the plane and the cylinder surfaces, respectively. This equation requires two main sets of boundary conditions. First of all, the Reynolds cavitation boundary condition requires the following:

$$p \geq 0 \text{ on } \Omega_c \text{ and } p = \frac{\partial P}{\partial x} = 0 \text{ on cavitation boundary} \tag{4}$$

These conditions are met by applying the penalty method proposed by Wu [23], which consists of adding a penalty term to the Reynolds equation, which becomes:

$$-\frac{\partial}{\partial X}\left(\overline{\varepsilon}\frac{\partial P}{\partial X}\right) + \frac{\partial(\overline{\rho}^*H)}{\partial X} + \underbrace{\xi P^-}_{\text{Penalty Term}} = 0 \tag{5}$$

where $P^-$ is defined as $P^- = P\theta(-P)$ where $\theta$—known as the Heaviside function—is nil for a positive value of $P$ and equal to unity for a negative one. Provided that the arbitrary constant $\xi$ has a sufficiently large value, the penalty term dominates the rest of the equation whenever $P^-$ has a value different than zero. Therefore, for any negative pressure value, the penalty term will dominate the equation, forcing the value of the pressure towards zero. The second set of boundary conditions is applied by setting both inlet and outlet pressures to nil.

It is important to note that viscosity and density vary with both pressure and temperature. On the other hand, viscosity also varies with shear stress, i.e., $\overline{\rho} = \overline{\rho}(P, \overline{T})$ and $\overline{\eta} = \overline{\eta}(P, \overline{T}, \overline{\tau})$. The term $\overline{\tau} = \|\tau/\tau_0\|$ is the dimensionless shear stress determined using the following equation:

$$\frac{Ha^2\tau_0}{\eta_R R}\int_0^1 \frac{\overline{\tau}_{zx}}{\overline{\eta}(P,\overline{T},\overline{\tau})}dZ = u_2 - u_1 \quad \text{with}: \quad \overline{\tau}_{zx} = \overline{\tau}_{zx}^0 + \frac{Hap_h}{R\tau_0}Z\frac{\partial P}{\partial X} \tag{6}$$

where $\overline{\tau}_{zx}^0$ is the dimensionless lubricant shear stress on the plane's surface ($Z = 0$). The last variable to be developed in Reynolds' equation (Equation (3)) is the dimensionless film thickness $H$, which is defined as follows:

$$H(X) = H_0 + \frac{X^2}{2} - W(X) \tag{7}$$

where $H_0$ is the rigid body separation term and $W$ is the dimensionless elastic deformation of the solids in the $z$-direction, in the contact zone $\Omega_c$. It is important to note that the deflections are obtained by assuming that the plane is rigid whereas the cylinder is elastic and accounts for both deformations. This is achieved by applying equivalent solid properties $(E, v)$ to the cylinder, where $E$ was defined in Equation (1) and $v = 0$. Then, the compound elastic deflection of both components is obtained using a classical linear elasticity approach. The governing equations after simplification are reduced to:

$$\begin{aligned} -\frac{\partial^2 U}{\partial X^2} - \frac{\partial}{\partial Z}\left[\frac{1}{2}\left(\frac{\partial U}{\partial Z} + \frac{\partial W}{\partial X}\right)\right] &= 0 \\ -\frac{\partial}{\partial X}\left[\frac{1}{2}\left(\frac{\partial U}{\partial Z} + \frac{\partial W}{\partial X}\right)\right] - \frac{\partial^2 W}{\partial Z^2} &= 0 \end{aligned} \tag{8}$$

where $U$ and $W$ are the total dimensionless elastic deflections in the $x$- and $z$-directions, respectively. In terms of boundary conditions for the linear elasticity equation, a zero-displacement boundary condition is applied to the boundary $\partial\Omega_b$. In addition, there is no tangential stress on the contact boundary (i.e., $\sigma_t = 0$ over $\Omega_c$). On the other hand, normal stress $\sigma_n$ is applied over the contact domain due to the hydrodynamic pressure applied, which can be represented as follows:

$$\frac{\partial W}{\partial Z} = -\frac{P}{2} \tag{9}$$

For the remaining boundaries, a free displacement boundary condition is applied ($\sigma_n = \sigma_t = 0$). Next, to consider the equilibrium of forces between the external load and the

pressure within the lubricating film—this will be achieved by monitoring the value of $H_0$ as discussed in later sections—a load balance equation is introduced as follows:

$$\int_{\Omega_c} P\,dX = \frac{\pi}{2} \tag{10}$$

On the other hand, the governing equations considering the thermal effects describe the heat generation and dissipation between the solids and the lubricant in both the *x*- and *z*-directions since the length in the *y*-direction is considered infinite. Therefore, the computational domain is formed by three consecutive rectangles shown in Figure 3 where the upper one is for the cylinder ($\Omega_2$), the one in the middle is for the fluid ($\Omega_f$), and the lower one is for the plane ($\Omega_1$). Concerning the depth required to ensure a zero temperature gradient within the depth of the solid domains, Kaneta et al. [34] and Wang et al. [35] proposed that a dimensionless depth of 3.15 is enough. In this paper, the dimensionless depth of the solid domains is taken to be 3.5. On the other hand, since the dimensionless depth is the ratio of the depth over the film thickness for the lubricant film as seen in Equation (2), it has a value of unity.

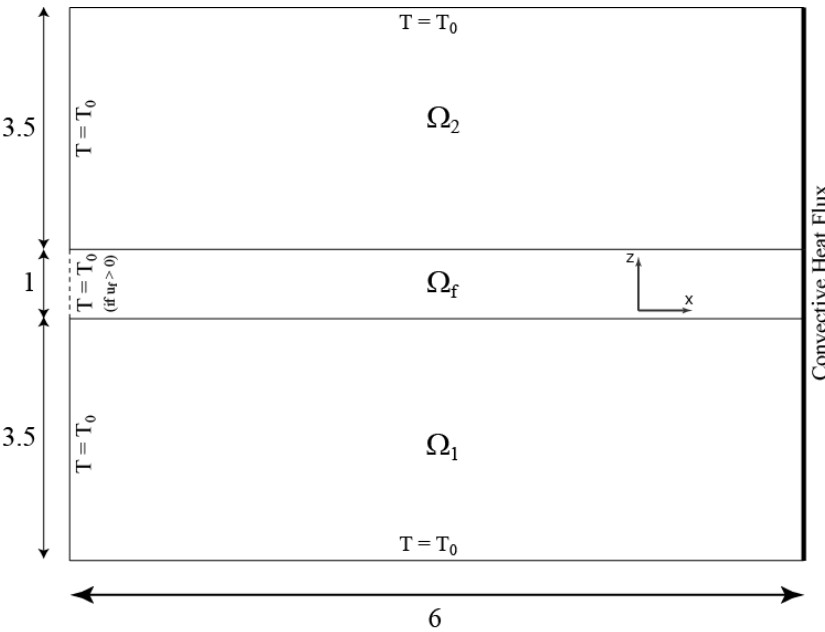

**Figure 3.** Computational domain of a line contact (Thermal Part).

By manipulating the energy equation and simplifying several terms, the equation governing the heat flow within the solids and the lubricant is given as follows:

$$\begin{cases} -\frac{\partial}{\partial X}\left(\frac{k_1}{a}\frac{\partial \overline{T}}{\partial X}\right) - \frac{\partial}{\partial Z}\left(\frac{k_1}{a}\frac{\partial \overline{T}}{\partial Z}\right) + \rho_1 c_1 u_1 \frac{\partial \overline{T}}{\partial X} = 0 & \text{(Solid p)} \\[2mm] -\frac{\partial}{\partial X}\left(\frac{k_2}{a}\frac{\partial \overline{T}}{\partial X}\right) - \frac{\partial}{\partial Z}\left(\frac{k_2}{a}\frac{\partial \overline{T}}{\partial Z}\right) + \rho_2 c_2 u_2 \frac{\partial \overline{T}}{\partial X} = 0 & \text{(Solid s)} \\[2mm] -\frac{\partial}{\partial X}\left(\frac{kR}{R}\frac{\partial \overline{T}}{\partial X}\right) - \frac{\partial}{\partial Z}\left(\frac{kR}{Ha^2}\frac{\partial \overline{T}}{\partial Z}\right) + \rho_R \overline{\rho} c\, u_f \frac{Ha}{R}\frac{\partial \overline{T}}{\partial X} = Q_{comp} + Q_{shear} & \text{(Lubricant Film)} \\[2mm] \text{With : } Q_{comp} = \frac{-Ha\,p_h}{RT_0}\frac{\overline{T}}{\overline{\rho}}\frac{\partial \overline{\rho}}{\partial \overline{T}} u_f \frac{\partial P}{\partial X} \text{ and } Q_{shear} = \frac{\overline{\eta}\,\eta_R Ha^2}{RT_0}\dot{\gamma}^2_{zx} \end{cases} \tag{11}$$

where $\rho$, $c$, and $k$ denote the density, specific heat, and thermal conductivity, respectively, while $Q_{comp}$ and $Q_{shear}$ correspond to the heat generation within the lubricant film by compression and by shear, respectively. The velocity of the lubricant in the *x*-direction $u_f$ can be computed as follows:

$$u_f = u_p + \frac{p_h H^2 a^3}{R^2 \eta_R}\frac{\partial P}{\partial X}\left[\int_0^Z \frac{Z\prime dZ\prime}{\overline{\eta}} - \frac{\overline{\eta}_e}{\overline{\eta}\prime_e}\int_0^Z \frac{dZ}{\overline{\eta}}\right] + \overline{\eta}_e(u_2 - u_1)\int_0^Z \frac{dZ}{\overline{\eta}} \tag{12}$$

In addition, $\dot{\gamma}_{zx}$—which is the *x*-component of the lubricant shear rate through the film thickness—used to compute $Q_{shear}$ can be defined as follows:

$$\dot{\gamma}_{zx} = \frac{\partial u_f}{\partial z} = \frac{Hap_h}{\eta_R \overline{\eta} R} \frac{\partial P}{\partial X} \left( Z - \frac{\overline{\eta}_e}{\overline{\eta}'_e} \right) + \frac{R}{Ha^2} \frac{\overline{\eta}_e}{\overline{\eta}} (u_2 - u_1) \tag{13}$$

The boundary conditions applied can be seen in Figure 3. Since the energy equation is hyperbolic, all the inlets have a temperature of $T_0$ ($\overline{T} = 1$). For the lubricant, $u_f$ can be negative in the proposed inlet region due to reverse flows (Poiseuille components). Accordingly, the ambient temperature boundary condition is only necessary for the inlet region $\left( u_f \geq 0 \right)$. For the solids, on the other hand, the left boundaries are actually inlet boundaries since both $u_2$ and $u_1$ are assumed to be positive throughout this paper. In addition, an ambient temperature is imposed on the top and bottom boundaries ($Z = -3.5$ and $Z = 4.5 \ \forall X$). As for the outlet boundaries, a convective heat flux boundary condition is assumed for both solids and the lubricant film by making the conductive heat flux nil ($kT_0/a \times \partial \overline{T}/\partial X = 0$). Lastly, for the two fluid–solid interfaces, a heat flux continuity is imposed using the following equations:

$$\frac{k_1}{a} \frac{\partial \overline{T}}{\partial Z} \bigg|_{Z=0^-} = \frac{kR}{Ha^2} \frac{\partial \overline{T}}{\partial Z} \bigg|_{Z=0^+} \quad \text{and} \quad \frac{k_2}{a} \frac{\partial \overline{T}}{\partial Z} \bigg|_{Z=1^+} = \frac{kR}{Ha^2} \frac{\partial \overline{T}}{\partial Z} \bigg|_{Z=1^-} \tag{14}$$

### 2.2. Circular Contact

For circular contact, the solid computational domain becomes a cube, which should have a sufficient dimensionless side length to make the half-space assumption valid. In this paper, a dimensionless side length of 60 is adopted, with the contact domain $\Omega_c$ located over the upper surface of the cube ($-4.5 \leq X \leq 1.5$ and $-3 \leq Y \leq 3$) as proposed by [21]. The symmetry of the problem is taken into consideration to reduce computational efforts, this is why only half of the computational domain is shown in Figure 4. The solution over the other half is deduced by symmetry.

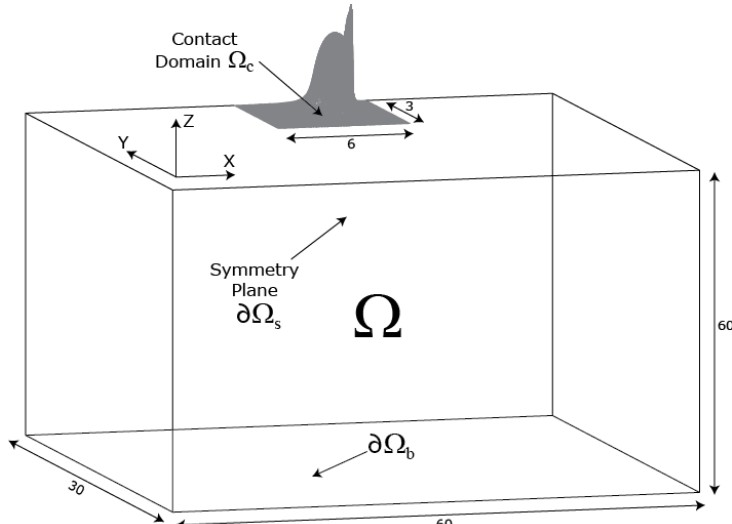

**Figure 4.** Computational domain for a circular contact (EHL part).

Reynolds equation is applied to the 2D contact domain $\Omega_c$ and it reads:

$$-\frac{\partial}{\partial X} \left( \overline{\varepsilon} \frac{\partial P}{\partial X} \right) - \frac{\partial}{\partial Y} \left( \overline{\varepsilon} \frac{\partial P}{\partial Y} \right) + \frac{\partial (\overline{\rho}^* H)}{\partial X} + \underbrace{\xi \, P^-}_{\text{Penalty Term}} = 0 \tag{15}$$

It can be seen from Equation (15) that the same penalty method proposed by Wu [23] is applied. In addition, similarly to the line contact case, a zero-pressure boundary condition

is to be applied over the boundaries of the contact domain $\Omega_c$ except for the symmetry boundary $\partial\Omega_{cs} = \Omega_c \cap \partial\Omega_s$ (where $\partial\Omega_s$ is the symmetry boundary of the solid domain), where a specific symmetry boundary condition should be imposed (i.e., $\partial P/\partial Y = 0$). The shear-dependence of viscosity within the Reynolds equation implies the need for the following shear stress equations to determine the shear stress profile across the lubricant film:

$$\int_0^1 \frac{\frac{Hap_h}{R}Z\frac{\partial P}{\partial X} + \overline{\tau}^0_{zx}p_h}{\eta_R\overline{\eta}}\frac{Ha^2}{R}dZ = u_2 - u_1$$
$$\int_0^1 \frac{\frac{Hap_h}{R}Z\frac{\partial P}{\partial X} + \overline{\tau}^0_{zy}p_h}{\eta_R\overline{\eta}}\frac{Ha^2}{R}dZ = 0 \tag{16}$$

The dimensionless shear stress components $\{\overline{\tau}_{zx}, \overline{\tau}_{zy}\}$ in the $x$- and $y$-directions can be accordingly computed as follows:

$$\overline{\tau}_{zx} = \overline{\tau}^0_{zx} + \frac{Ha}{R}Z\frac{\partial P}{\partial X}$$
$$\overline{\tau}_{zy} = \overline{\tau}^0_{zy} + \frac{Ha}{R}Z\frac{\partial P}{\partial Y} \tag{17}$$

where $\overline{\tau}^0_{zx}$ and $\overline{\tau}^0_{zy}$ are the shear stress $x$- and $y$-components over the plane surface. On the other hand, the film thickness equation becomes:

$$H(X,Y) = H_0 + \frac{X^2 + Y^2}{2} - W(X,Y) \tag{18}$$

The linear elasticity equations will govern the deformations $U$, $V$, and $W$, which are in the $x$-, $y$-, and $z$-directions, respectively, as follows:

$$\begin{cases} -\frac{\partial^2 U}{\partial X^2} - \frac{\partial}{\partial Y}\left[\frac{1}{2}\left(\frac{\partial U}{\partial Y} + \frac{\partial V}{\partial X}\right)\right] - \frac{\partial}{\partial Z}\left[\frac{1}{2}\left(\frac{\partial U}{\partial Z} + \frac{\partial W}{\partial X}\right)\right] = 0 \\ -\frac{\partial}{\partial X}\left[\frac{1}{2}\left(\frac{\partial U}{\partial Y} + \frac{\partial V}{\partial X}\right)\right] - \frac{\partial^2 V}{\partial Y^2} - \frac{\partial}{\partial Z}\left[\frac{1}{2}\left(\frac{\partial V}{\partial Z} + \frac{\partial W}{\partial Y}\right)\right] = 0 \\ -\frac{\partial}{\partial X}\left[\frac{1}{2}\left(\frac{\partial U}{\partial Z} + \frac{\partial W}{\partial X}\right)\right] - \frac{\partial}{\partial Y}\left[\frac{1}{2}\left(\frac{\partial V}{\partial Z} + \frac{\partial W}{\partial Y}\right)\right] - \frac{\partial^2 W}{\partial Z^2} = 0 \end{cases} \tag{19}$$

Boundary conditions should be applied in order to complete Equation (19) as follows:

$$\begin{array}{ll} \frac{\partial W}{\partial Z} = \frac{-2P}{\pi} \text{ and } \{\sigma_t\} = \{\varnothing\} & \text{over } \Omega_c \\ U = V = W = 0 & \text{over } \partial\Omega_b \\ V = 0 \text{ and } \{\sigma_t\} = \{\varnothing\} & \text{over } \partial\Omega_s \\ \sigma_n = 0 \text{ and } \{\sigma_t\} = \{\varnothing\} & \text{elsewhere} \end{array} \tag{20}$$

Lastly, the load balance equation for circular contacts can be written as:

$$\int_{\Omega_c} P\,dXdY = \frac{\pi}{3} \tag{21}$$

On the other hand, the computational domain of the thermal part is now three rectangular cuboids as shown in Figure 5. Similar to the line contact case, a dimensionless depth of 3.5 is adopted for the solid domains to ensure a zero temperature gradient within their depth.

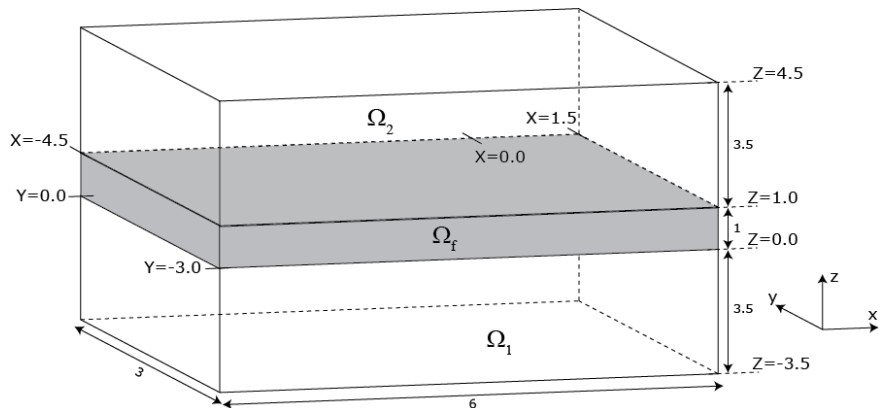

**Figure 5.** Computational Domain for a circular contact (Thermal).

For point contacts, the energy equations after simplification become:

$$
\begin{cases}
-\frac{\partial}{\partial X}\left(\frac{k_1}{a}\frac{\partial \overline{T}}{\partial X}\right) - \frac{\partial}{\partial Y}\left(\frac{k_1}{a}\frac{\partial \overline{T}}{\partial Y}\right) - \frac{\partial}{\partial Z}\left(\frac{k_1}{a}\frac{\partial \overline{T}}{\partial Z}\right) + \rho_1 c_1 u_1 \frac{\partial \overline{T}}{\partial X} = 0 \\[4pt]
-\frac{\partial}{\partial X}\left(\frac{k_2}{a}\frac{\partial \overline{T}}{\partial X}\right) - \frac{\partial}{\partial Y}\left(\frac{k_2}{a}\frac{\partial \overline{T}}{\partial Y}\right) - \frac{\partial}{\partial Z}\left(\frac{k_2}{a}\frac{\partial \overline{T}}{\partial Z}\right) + \rho_2 c_2 u_2 \frac{\partial \overline{T}}{\partial X} = 0 \\[4pt]
-\frac{\partial}{\partial X}\left(\frac{kH}{R}\frac{\partial \overline{T}}{\partial X}\right) - \frac{\partial}{\partial Y}\left(\frac{kH}{a}\frac{\partial \overline{T}}{\partial Y}\right) - \frac{\partial}{\partial Z}\left(\frac{kR}{Ha^2}\frac{\partial \overline{T}}{\partial Z}\right) + \rho_R \overline{\rho} c \frac{Ha}{R}\left(u_f \frac{\partial \overline{T}}{\partial X} + v_f \frac{\partial \overline{T}}{\partial Y}\right) = Q_{comp} + Q_{shear} \\[4pt]
\text{Where}: Q_{comp} = \frac{\overline{T}}{\overline{\rho}}\frac{p_h Ha}{T_0 R}\frac{\partial \overline{\rho}}{\partial \overline{T}}\left(u_f \frac{\partial P}{\partial X} + v_f \frac{\partial P}{\partial Y}\right) \text{ and } Q_{shear} = \frac{\eta_R \overline{\eta}}{T_0}\frac{Ha^2}{R}\left(\dot{\gamma}^2_{zx} + \dot{\gamma}^2_{zy}\right)
\end{cases}
\tag{22}
$$

where $u_f$ can be calculated using Equation (12), and $v_f$ can be computed as follows:

$$
v_f = v_c + \frac{p_h H^2 a^3}{R^2 \eta_R}\frac{\partial P}{\partial Y}\left[\int_0^Z \frac{Z' dZ'}{\overline{\eta}} - \frac{\overline{\eta}_e}{\overline{\eta}'_e}\int_0^Z \frac{dZ}{\overline{\eta}}\right]
\tag{23}
$$

Moreover, $\dot{\gamma}_{zx}$ can be calculated using Equation (13), whereas $\dot{\gamma}_{zy}$ can be calculated as:

$$
\dot{\gamma}_{zy} = \frac{\partial v_f}{\partial z} = \frac{Ha p_h}{\eta_R \overline{\eta} R}\frac{\partial P}{\partial Y}\left(Z - \frac{\overline{\eta}_e}{\overline{\eta}'_e}\right)
\tag{24}
$$

To complete Equation (22), some boundary conditions should be imposed:

$$
\begin{aligned}
\overline{T} &= 1 && \text{over the inlet boundaries of both solid and fluid domains} \\
\overline{T} &= 1 && \text{over the depth of the solid domains} \\
\nabla \overline{T} \cdot \vec{n} &= 0 && \text{over outlet and symmetry boundaries of solid and fluid domains}
\end{aligned}
\tag{25}
$$

where the depth of solid domains can be represented by $Z = -3.5$ and $Z = 4.5\ \forall X$ and $Y$, the outlet boundaries can be represented by $X = 1.5\ \forall Y$ and $Z$ in addition to $Y = -3\ \forall X$ and $Z$, and the symmetry boundaries can be represented as $Y = 0\ \forall X$ and $Z$. It is also important to note that, similar to the line contact case, for the fluid domain inlet boundary (i.e., $X = -4.5$ and $0 \leq Z \leq 1\ \forall Y$), $\overline{T} = 1$ should only be imposed on the portion where $u_f > 0$. In fact, the inlet and outlet of the fluid cannot be determined in advance due to the reverse flows arising from a strong Poiseuille component. Lastly, the continuity condition across the two fluid–solid interfaces ($Z = 0$ and $Z = 1$) is taken into account using Equation (14), similar to the line contact case.

## 3. FEM Full Model

The physical problem described in Section 2 entails a total of five field variables: $P$ for the pressure distribution over the contact domain, $\overline{\tau}^0 (= \overline{\tau}^0_{zx}$ for line contacts, or $\{\overline{\tau}^0_{zx}, \overline{\tau}^0_{zy}\}$ for circular contacts) for the shear stress distribution over the flat plane surface,

$\overline{U}(= \{U, V\}$ for line contacts, or $\{U, V, W\}$ for circular contacts) for the elastic deformation over the equivalent solid domain, $H_0$ for the rigid body separation term, and $\overline{T}$ for the temperature variation over the solid and fluid domains. These field variables will be solved for using Equations (5), (6), (8), (10) and (11), respectively, for line contacts and using Equations (15), (16), (19), (21) and (22) for circular contacts. These equations for both line and circular contacts can be written in matrix form using subscript $e, h, l, t, s$ accordingly for the elastic, hydrodynamic, load balance, thermal, and shear stress domains. All equations are discretized using non-regular non-structured finite elements (i.e., triangular elements for line contacts and tetrahedral elements for circular contacts), except for the fluid domain of the thermal part where structured elements are employed (i.e., rectangular elements for line contacts and prism elements for circular contacts). Second-order Lagrange elements are used for all field variables (except for $H_0$, which is a scalar). Details of the corresponding finite element formulations will not be provided here, in the interest of space. Interested readers are referred to [20]. All equations are solved simultaneously (i.e., as a monolithic system), and since the resulting system of equations is highly non-linear, it is solved using a damped-Newton procedure [36], which results in a linearized system of equations at every Newton iteration $k$ of the form:

$$
\begin{bmatrix}
K_{ee} & K_{eh} & \varnothing & \varnothing & \varnothing \\
\overline{K}_{he} & \overline{K}_{hh} & \overline{K}_{hl} & \overline{K}_{ht} & \overline{K}_{hs} \\
\varnothing & \overline{K}_{lh} & 0 & \varnothing & \varnothing \\
\overline{K}_{te} & \overline{K}_{th} & \overline{K}_{tl} & \overline{K}_{tt} & \overline{K}_{ts} \\
\overline{K}_{se} & \overline{K}_{sh} & \overline{K}_{sl} & \overline{K}_{st} & \overline{K}_{ss}
\end{bmatrix}^{k-1}
\begin{Bmatrix}
\delta \overline{U} \\
\delta \overline{P} \\
\delta \overline{H}_0 \\
\delta \overline{T} \\
\delta \overline{\tau}^0
\end{Bmatrix}^{k}
= -
\begin{Bmatrix}
\varnothing \\
\overline{R}_h \\
\overline{R}_l \\
\overline{R}_t \\
\overline{R}_s
\end{Bmatrix}^{k-1}
\qquad (26)
$$

The matrix on the left represents the Jacobian matrix of the non-linear TEHD problem, while the right-hand-side vector is the residual vector. The vector of unknowns consists of the increments of the field variables $\delta\overline{U}, \delta P, \delta H_0, \delta\overline{T}$, and $\delta\overline{\tau}^0$. At every iteration $k$, it is to be multiplied by a damping factor and then added to the overall solution vector of the previous iteration $k - 1$. It is important to note that the Jacobian matrix should be computed for every iteration $k$ with the exception of $K_{ee}$, $K_{eh}$, and $K_{lh}$, which are only computed once (at the first iteration) since both linear elasticity and load balance equations are linear. Using the subscripts mentioned above, the total number of dofs for line and circular contacts can be defined as $n_{dof} = 2 \times n_e + n_h + 1 + n_t + n_s$ and $n_{dof} = 3 \times n_e + n_h + 1 + n_t + 2 \times n_s$ respectively, where $n$ refers to the number of nodes in a specific domain. The main drawback of solving the "full model" described in Equation (26) is that several dofs in the elastic domain are solved in vain. In fact, only normal elastic deflections over the contact domain $\Omega_c$ are needed. The purpose of this paper is to investigate how the proposed reduced model can help in decreasing the computational overhead by decreasing the number of elastic dofs calculated by $2 \times n_e - n_h$ for linear contact cases, and $3 \times n_e - n_h$ for circular contacts (i.e., by reducing the number of elastic dofs to $n_h$ in both cases). The description of the reduction technique is detailed in the following section.

## 4. Model Order Reduction

The following section describes the model order reduction technique used to decrease the number of dofs by eliminating unneeded ones from the elastic domain. This is performed using static condensation in addition to a splitting procedure. This section will be divided into 3 sub-sections where the static condensation, the splitting technique, and their application to the actual finite element model are detailed.

### 4.1. Static Condensation

The static condensation technique, also known as the Schur complement method, or Guyan condensation was long used in finite element structural analysis. This method became well-known after being introduced by Guyan [29] and Irons [37]. To detail how

the procedure is carried out, a linear elasticity problem governed by the following overall matrix system is considered:

$$[K]\{U\} = \{F\} \tag{27}$$

where $[K]$ is the stiffness matrix of the structure, $\{U\}$ is the vector of nodal displacements, and $\{F\}$ is the vector of external nodal forces. The static condensation technique consists of separating needed dofs (named masters with subscript $m$) from unneeded ones (named slaves with subscript $s$). Equation (27) becomes the following after re-arrangement:

$$\begin{bmatrix} K_{ss} & K_{sm} \\ \hline K_{ms} & K_{mm} \end{bmatrix} \left\{ \frac{U_s}{U_m} \right\} = \left\{ \frac{F_s}{F_m} \right\} \tag{28}$$

Expanding the matrix above to two separate equations and using the upper one to express $\{U_s\}$ as a function of $\{U_m\}$ will lead to the following:

$$\{U_s\} = -[K_{ss}]^{-1}[K_{sm}]\{U_m\} + [K_{ss}]^{-1}\{F_s\} \tag{29}$$

Readjusting the lower part of Equation (28) using Equation (29) to express $\{U_s\}$ as a function of $\{U_m\}$ will help in eliminating the slaves from the overall matrix. The lower part of Equation (28) becomes:

$$[\hat{K}]\{U_m\} = \{\hat{F}\}$$
$$\text{with}: \ [\hat{K}] = [K_{mm}] - [K_{ms}][K_{ss}]^{-1}[K_{sm}] \text{ and } \{\hat{F}\} = \{F_m\} - [K_{ms}][K_{ss}]^{-1}\{F_s\} \tag{30}$$

As can be seen in Equation (30), which is known as the reduced system, even though slaves were eliminated, their effect is injected in both stiffness and force matrices keeping the solution exact. In fact, static condensation mainly differs from other MOR techniques in that no additional approximations are introduced, beyond discretization ones. In addition, it can also be seen that this method only requires a basic knowledge of linear algebra, which makes it easy to implement even for novice users. Lastly, it is important to note that the main drawback of this method is that the reduced stiffness matrix obtained $[\hat{K}]$ is a dense one compared to sparse matrices emerging from standard finite element formulations. Accordingly, a splitting procedure is required to alleviate this shortcoming, as discussed in the following section.

### 4.2. Splitting Procedure

The main purpose of the splitting procedure is to retrieve the standard sparsity of the overall matrix system after static condensation is applied. In order to do so, the same matrix system shown in Equation (27) is used for demonstration purposes. The first step would be to split the matrix $[K]$ into two parts, $[K^n]$ and $[K^f]$, such that $[K] = [K^n] + [K^f]$. The next step would be to set an initial guess $\{U\}^0$ for the solution at the first iteration ($i = 1$) and apply an iterative procedure as follows:

$$[K^n]\{U\}^i = \{F\} - [K^f]\{U\}^{i-1} \tag{31}$$

where $\{U\}^i$ is the new solution vector for every iteration $i$. In addition, the choice of $[K^n]$ should be made in such a way as to reduce the computational overhead as much as possible (e.g., diagonal or band matrix). The choice adopted in this paper will be elaborated on in Section 4.3.

### 4.3. Overall Numerical Procedure

This section describes how the static condensation and the splitting procedure introduced earlier are applied to the full model described in Equation (26). First of all, the master dofs are chosen to be the $W$ deformations across the contact domain $\Omega_c$ with a number of $n_m = n_h$. Accordingly, for line contacts, the nodal dofs that are not needed

from the elastic field are the $U$ component belonging to the contact domain, and both $U$ and $V$ components for the nodes of $\Omega - \Omega_c$, and their total number is $n_s = 2 \times n_e - n_h$. On the other hand, this number is larger for circular contacts since both $U$ and $V$ components are not needed over $\Omega_c$ whereas $U$, $V$, and $W$ components are not needed over $\Omega - \Omega_c$, meaning that $n_s = 3 \times n_e - n_h$. Using static condensation defined in Section 4.1, Equation (26) is modified as follows:

$$
\begin{bmatrix}
\hat{K}_{ee} & \hat{K}_{eh} & \varnothing & \varnothing & \varnothing \\
\bar{K}_{he} & \bar{K}_{hh} & \bar{K}_{hl} & \bar{K}_{ht} & \bar{K}_{hs} \\
\varnothing & \bar{K}_{lh} & 0 & \varnothing & \varnothing \\
\hat{K}_{te} & \bar{K}_{th} & \bar{K}_{tl} & \bar{K}_{tt} & \bar{K}_{ts} \\
\hat{K}_{se} & \bar{K}_{sh} & K_{sl} & K_{st} & K_{ss}
\end{bmatrix}^{k-1}
\begin{Bmatrix}
\delta \hat{W} \\
\delta \bar{P} \\
\delta \bar{H}_0 \\
\delta \bar{T} \\
\delta \bar{\tau}^0
\end{Bmatrix}^{k}
= -
\begin{Bmatrix}
\varnothing \\
\bar{R}_h \\
\bar{R}_l \\
\bar{R}_t \\
\bar{R}_s
\end{Bmatrix}^{k-1}
\tag{32}
$$

where $\delta \hat{W}$ corresponds to the elastic deformation increment in the $z$-direction over the contact domain $\Omega_c$. It can be also referred to as the "Reduced elastic domain". It is also important to note that there are no forces that are applied to the slaves since the force is only applied to the contact nodes chosen as masters. Accordingly, $\{F_s\} = \{\varnothing\}$ and $\{\hat{F}\} = \{F_m\}$ using Equation (30), meaning that $[\hat{K}_{eh}]$ is nothing but $[K_{eh}]$ with the zero lines removed. Similarly, $[\hat{K}_{he}]$ is nothing but $[K_{he}]$ with the zero columns removed since the hydrodynamic problem is only connected to the elastic domain through the master dofs (in a finite element sense). For the same reason, $[\hat{K}_{te}]$ and $[\hat{K}_{se}]$ are nothing but $[K_{te}]$ and $[K_{se}]$, respectively, with the zero columns removed. Lastly, using Equation (30), $[\hat{K}_{ee}]$ can be defined as:

$$
[\hat{K}_{ee}] = [K_{mm}] - [K_{ms}][K_{ss}]^{-1}[K_{sm}]
\tag{33}
$$

To optimize the computational overhead, the matrix $[K_{ss}]$ is not inverted. Instead, $\left[\widetilde{K}\right] = [K_{ss}]^{-1}[K_{sm}]$ is evaluated by solving the following system of equations for $\left[\widetilde{K}\right]$:

$$
[K_{ss}]\left[\widetilde{K}\right] = [K_{sm}]
\tag{34}
$$

The direct solver UMFPACK [38] is used to generate the LU decomposition of $[K_{ss}]$. It is important to note that even though this decomposition is time-consuming, it is performed only once, whereas the computationally cheap operations of backward and forward substitutions are repeated for every column of $[K_{sm}]$ to obtain its corresponding column within $\left[\widetilde{K}\right]$. The next step would be to compute $[\hat{K}_{ee}]$ using Equation (33). From here on, the evaluation of $[\hat{K}_{ee}]$ will be referred to as the offline phase of the MOR technique. One of the main advantages of this method is that the offline phase can only be performed once for a given mesh, and then stored for later use. In fact, $[\hat{K}_{ee}]$ is independent of operating conditions, and through the adopted choice of equivalent material properties, it was also made independent of the latter (see Equations (8) and (19)). However, even though $[\hat{K}_{ee}]$ is much smaller in size than $[K_{ee}]$, it may contain more non-zeros, since it is dense [28]. Accordingly, the splitting technique is used where $[\hat{K}_{ee}]$ is divided into "near" and "far" contributions named $\left[\hat{K}_{ee}^{n}\right]$ and $\left[\hat{K}_{ee}^{f}\right]$, respectively, with $[\hat{K}_{ee}] = \left[\hat{K}_{ee}^{n}\right] + \left[\hat{K}_{ee}^{f}\right]$. The choice of near and far contributions was made similarly to previous works on numerical EHL such as [39,40]. In fact, for a given node, the near contributions were considered to come from the nodes within the same element(s) (including the node itself), whereas the rest are considered as far. Therefore, for every row of $[\hat{K}_{ee}]$, the contributions from nodes in the same

element are shifted towards $\left[\hat{K}_{ee}^{n}\right]$ and the rest are shifted towards $\left[\hat{K}_{ee}^{f}\right]$. Consequently, Equation (32) becomes:

$$
\begin{bmatrix}
\hat{K}_{ee}^{n} & \hat{K}_{eh} & \varnothing & \varnothing & \varnothing \\
\hat{K}_{he} & \bar{K}_{hh} & \bar{K}_{hl} & \bar{K}_{ht} & \bar{K}_{hs} \\
\varnothing & \bar{K}_{lh} & 0 & \varnothing & \varnothing \\
\hat{K}_{te} & \bar{K}_{th} & \bar{K}_{tl} & \bar{K}_{tt} & \bar{K}_{ts} \\
\hat{K}_{se} & \bar{K}_{sh} & \bar{K}_{sl} & \bar{K}_{st} & \bar{K}_{ss}
\end{bmatrix}^{k-1}
\begin{Bmatrix}
\delta\hat{W}^{i} \\
\delta P^{i} \\
\delta H_{0}^{i} \\
\delta\bar{T}^{i} \\
\delta\bar{\tau}^{0i}
\end{Bmatrix}^{k}
= -
\begin{Bmatrix}
\hat{K}_{ee}^{f}\delta\hat{W}^{i-1} \\
\bar{R}_{h} \\
\bar{R}_{l} \\
\bar{R}_{t} \\
\bar{R}_{s}
\end{Bmatrix}^{k-1}
\tag{35}
$$

where $i$ is the iteration number for the splitting procedure (internal loop within every damped-Newton iteration $k$). To start with, a specific initial guess is defined where the Hertzian contact pressure is used for $P$, its corresponding reduced elastic deformation field for $\hat{W}$, a carefully chosen value for $H_0$, a dimensionless value of unity was chosen for temperature, and the shear stress initial guess is chosen to be nil. Note that the matrix system in Equation (35) is repeatedly solved for every iteration $k$ where the overall increment vector $\left\{\delta\hat{W}, \delta P, \delta H_0, \delta\bar{T}, \delta\bar{\tau}^0\right\}$ is multiplied by a damping factor, and then added to the solution obtained at the previous Newton iteration. These iterations—using a damped Newton procedure—will be stopped according to the convergence criteria defined in [36]. It is also important to note that the splitting procedure also requires some iterations (denoted by the iteration index $i$), as an internal loop that is carried out within every damped-Newton iteration $k$. For every splitting iteration $i$, the far contribution of the elastic residual vector is evaluated on the right-hand side using the reduced elastic deformation vector computed at the previous splitting iteration $i-1$. Within the splitting procedure, the overall system of equations is repeatedly solved until the L2-norm (normalized with respect to the total number of unknowns) of the absolute difference between two consecutive iterations $i-1$ and $i$ of the increment vector of the overall solution $\left\{\delta\hat{W}, \delta P, \delta H_0, \delta\bar{T}, \delta\bar{\tau}^0\right\}$ falls below $10^{-5}$. Finally, it is important to note that the static condensation with splitting reduces the size of the problem from $n_{dof} = 2 \times n_e + n_h + 1 + n_t + n_s$ to $\hat{n}_{dof} = 2 \times n_h + 1 + n_t + n_s$ for line contacts and from $n_{dof} = 3 \times n_e + n_h + 1 + n_t + 2 \times n_s$ to $\hat{n}_{dof} = 2 \times n_h + 1 + n_t + 2 \times n_s$ for circular contacts. Even though the reduction is large, especially given that the elastic domain is one of the largest in the problem, it is not as significant as for other MOR techniques that are based on mode superposition principles [24–27,41]. On the other hand, the significance of this technique is that it is fast and efficient since the Jacobian matrix evaluation and its LU decomposition are not repeated at every splitting iteration $i$. The same applies to the hydrodynamic, load balance, thermal, and shear stress components of the right-hand-side residual vector, which are not re-evaluated at every splitting iteration. Therefore, only the evaluation of the elastic component of the residual vector and the forward and backward substitutions are carried out for every splitting iteration $i$, which are light in terms of computational overhead. In addition, the main significance of this method is its simplicity since, as already explained, the method only needs some basic knowledge in linear algebra in contrast to other MOR techniques, where a reduced solution space requiring a high level of expertise should be defined. Moreover, the MOR technique introduced is special in terms of its independence of both material properties and operating conditions. In fact, the reduced matrix is only computed once for a given mesh. However, this is not the case for other techniques where the reduced solution space should be re-defined every time the solid material properties are changed or a new feature is incorporated into the problem (e.g., surface roughness). Lastly, this technique reduces the actual EHL parts from 2D-1D and 3D-2D coupled problems to 1D-1D and 2D-2D for line and circular contacts, respectively. This will lead to significant savings in central processing unit (cpu) times, as will be discussed next.

### 5. Results and Discussion

In this section, the performance of the proposed MOR technique is studied, compared to the full model. A single Intel Core i7 2.7 GHz processor was used for all simulations. To complete the employed numerical model, lubricant viscosity dependence relations on temperature, pressure, and shear stress are required. First, the double-Newtonian modified Carreau model [42] is used to represent the shear-dependence of the dimensionless generalized Newtonian viscosity $\overline{\eta}$ as follows:

$$\overline{\eta}\left(P,\overline{T},\overline{\tau}\right) = \overline{\mu}_2\left(P,\overline{T}\right) + \frac{\overline{\mu}_1\left(P,\overline{T}\right) - \overline{\mu}_2\left(P,\overline{T}\right)}{\left[1 + \left(\frac{\overline{\tau}\,\tau_0}{G_c}\right)^{a_c}\right]^{\frac{\frac{1}{n_c}-1}{a_c}}} \tag{36}$$

where the choice of $\tau_0 = p_h$ is adopted. Further, $a_c$, $n_c$, and $G_c$ are constants. In addition, the Newtonian viscosities $\overline{\mu}_1$ and $\overline{\mu}_2$ in Equation (36) are dependent on both temperature and pressure. This correlation is described by the Roelands [43] equation for simplicity as follows:

$$\overline{\mu}_i\left(P,\overline{T}\right) = \frac{\mu_{i,R}}{\mu_{1,R}}\exp\left\{\left(\ln(\mu_{1,R}) + 9.67\right)\left[-1 + \left(1 + 5.1 \times 10^{-9}P\,p_h\right)^{Z_0}\left(\frac{\overline{T}\,T_R - 138}{T_R - 138}\right)^{-S_0}\right]\right\} \tag{37}$$

$$\text{With}: i = 1 \text{ or } 2, \quad Z_0 = \frac{\alpha}{[5.1 \times 10^{-9}(\ln(\mu_{1,R}) + 9.67)]} \text{ and } S_0 = \frac{\beta(T_R - 138)}{\ln(\mu_{1,R}) + 9.67}$$

where $\mu_{1,R}$ and $\mu_{2,R}$ are the first and second Newtonian viscosity limits at the reference state under zero and infinite shear rates, respectively; $\alpha$ is the lubricant pressure–viscosity coefficient and $\beta$ is the lubricant temperature–viscosity coefficient. Note that the reference state corresponds to $T_R = T_0 = 300K(\overline{T}_R = 1)$, where $T_0$ is the ambient temperature, $p_R = 0$ Pa, and $\tau_R = 0$ Pa. Moreover, the lubricant density dependence on both temperature and pressure is needed. For this, the Dowson and Higginson [44] relation is used for simplicity as follows:

$$\overline{\rho}\left(P,\overline{T}\right) = \frac{0.59 \times 10^9 + 1.34P\,p_h}{0.59 \times 10^9 + P\,p_h} - \gamma T_R\left(\overline{T} - 1\right) \tag{38}$$

where $\gamma$ is the lubricant temperature-density coefficient. Note that both the Roelands relation and the Dowson and Higginson equation of state do not accurately represent the pressure-temperature dependence of the viscosity and density of common lubricants. Nonetheless, given that the current work is purely numerical, with the objective of developing a fast and reliable numerical approach for the analysis of TEHL contacts, such relations are preferred for their simplicity. Moreover, they are well-known and commonly used in the TEHL literature. However, the reader is reminded that, when a real performance prediction of TEHL contacts is sought, more realistic—but also more sophisticated—rheological models are required, whether for the prediction of film thickness [45] or friction [46].

It is important to note that the performance of the proposed MOR technique will be studied for moderate and heavy loads. The solution of the latter is known to be numerically challenging, requiring the use of special stabilized FEM formulations [22] to remove inherent numerical instabilities/oscillations. Thus, the limiting shear stress (LSS) behavior is considered since it will be reached especially under high loads. This concept proposes the existence of an asymptotic shear stress value that is independent of the shear rate (i.e., any increase in shear rate does not lead to an increase in shear stress). This effect is incorporated as proposed by [47], i.e., the LSS $\tau_L$ is related to pressure as follows:

$$\tau_L = \Lambda\,p \tag{39}$$

where $\Lambda$ is a constant deduced from EHL experimental friction curves. This will only change the value of the shear stress when it exceeds the LSS by making it equal to $\tau_L$. Lastly, the considered material properties and operating conditions are summarized in Table 1.

**Table 1.** Lubricant properties, solid properties, and operating conditions.

| Material Properties | | Operating Conditions | |
|---|---|---|---|
| **Lubricant** | | $T_R = T_0 = 300$ K | |
| $\mu_{1,R} = 0.1$ Pa·s | $c = 1500$ J/kg·K | $R = 15$ mm | |
| $\mu_{2,R}/\mu_{1,R} = 0.5$ | $k = 0.1$ W/m·K | SRR = 0.0–0.5 | |
| $\alpha = 20$ GPa$^{-1}$ | $G_c = 0.01$ MPa | **Line Contacts** | |
| $\beta = 0.05$ K$^{-1}$ | $a_c = 2.2$ | **Moderate Load** | **Heavy Load** |
| $\gamma = 0.00075$ K$^{-1}$ | $n_c = 0.8$ | $u_m = 0.1$ m/s | $u_m = 0.5$ m/s |
| $\rho_R = 750$ kg/m$^3$ | $\Lambda = 0.075$ | $F = 0.2$ MN/m | $F = 2$ MN/m |
| **Solids** | | $(p_h = 0.7$ GPa$)$ | $(p_h = 2.2$ GPa$)$ |
| $\rho_1 = \rho_2 = 7850$ kg/m$^3$ | | **Point Contacts** | |
| $E_1 = E_2 = 210$ GPa | | **Moderate Load** | **Heavy Load** |
| $v_1 = v_2 = 0.3$ | | $u_m = 0.1$ kg/m$^3$ | $u_m = 0.5$ m/s |
| $k_1 = k_2 = 21$ W/m·K | | $F = 25$ N | $F = 1000$ N |
| $c_1 = c_2 = 470$ J/kg·K | | $(p_h = 0.66$ GPa$)$ | $(p_h = 2.25$ GPa$)$ |

*5.1. Line Contacts*

The results were computed using the "Normal" mesh defined in [20], which was shown to provide grid-independent solutions. The mesh corresponds to 2450 triangular elements used to discretize the 2D domain $\Omega$. The projection of these elements on the contact region was used in discretizing the 1D contact domain (to avoid mapping), resulting in a total of 249 line elements for $\Omega_c$. On the other hand, 2116 triangular elements were used to discretize the plane domain $\Omega_1$. The same number was used for the cylinder domain $\Omega_2$, of which mesh is nothing but a mirror image of that of the plane, with respect to the mid-layer of the lubricant film (Z = 0.5). Lastly, rectangular elements were used to discretize the fluid domain $\Omega_f$ of the thermal part, for which projection over the contact domain $\Omega_c$ is nothing but its corresponding 1D mesh. For the "Normal" mesh case, the total number of elements across the film thickness is 4. Accordingly, the mesh used has a total of 499 nodes/degrees of freedom (dofs) for the hydrodynamic and shear stress parts, 5195 nodes (10,390 dofs) for the elastic part, and 12,491 nodes/dofs for the thermal part, adding up to a total of $n_{\text{dof}} = 23,880$. Accordingly, the reduced number of degrees of freedom after static condensation is $\hat{n}_{\text{dof}} = 13,989$. Figure 6 shows the dimensionless pressure and film thickness distributions over the contact domain (top), temperature rise within the mid-layer of the lubricant film (middle), and dimensionless shear stress variations over the plane surface (Z = 0), for both considered loading conditions (moderate and high) with $SRR = 0.5$, obtained using both the full and reduced models. It can be seen in Figure 6 that the full and reduced model solutions are in perfect agreement. This is expected since the applied MOR method is exact, which means it does not include any approximations. Note that compared to the isothermal Newtonian case (not shown here), the film thicknesses reported in Figure 6 under thermal non-Newtonian considerations are lower due to the combined effects of inlet shear-thinning, as well as compressive and shear heating. These also reduce the lubricant shear stress across the film, compared to the isothermal Newtonian case. The effect on pressure would be minimal though, except in the vicinity of the pressure spike, which is usually thinner and higher in the isothermal Newtonian case.

The proposed methodology was also used to generate friction curves (i.e., friction coefficient $f$ vs. slide-to-roll ratio $SRR$), where:

$$f = \frac{\int_{\Omega_c} \tau_{zx}|_{z=h/2} d\Omega}{F} \quad \text{and} \quad SRR = \frac{u_2 - u_1}{u_m} \tag{40}$$

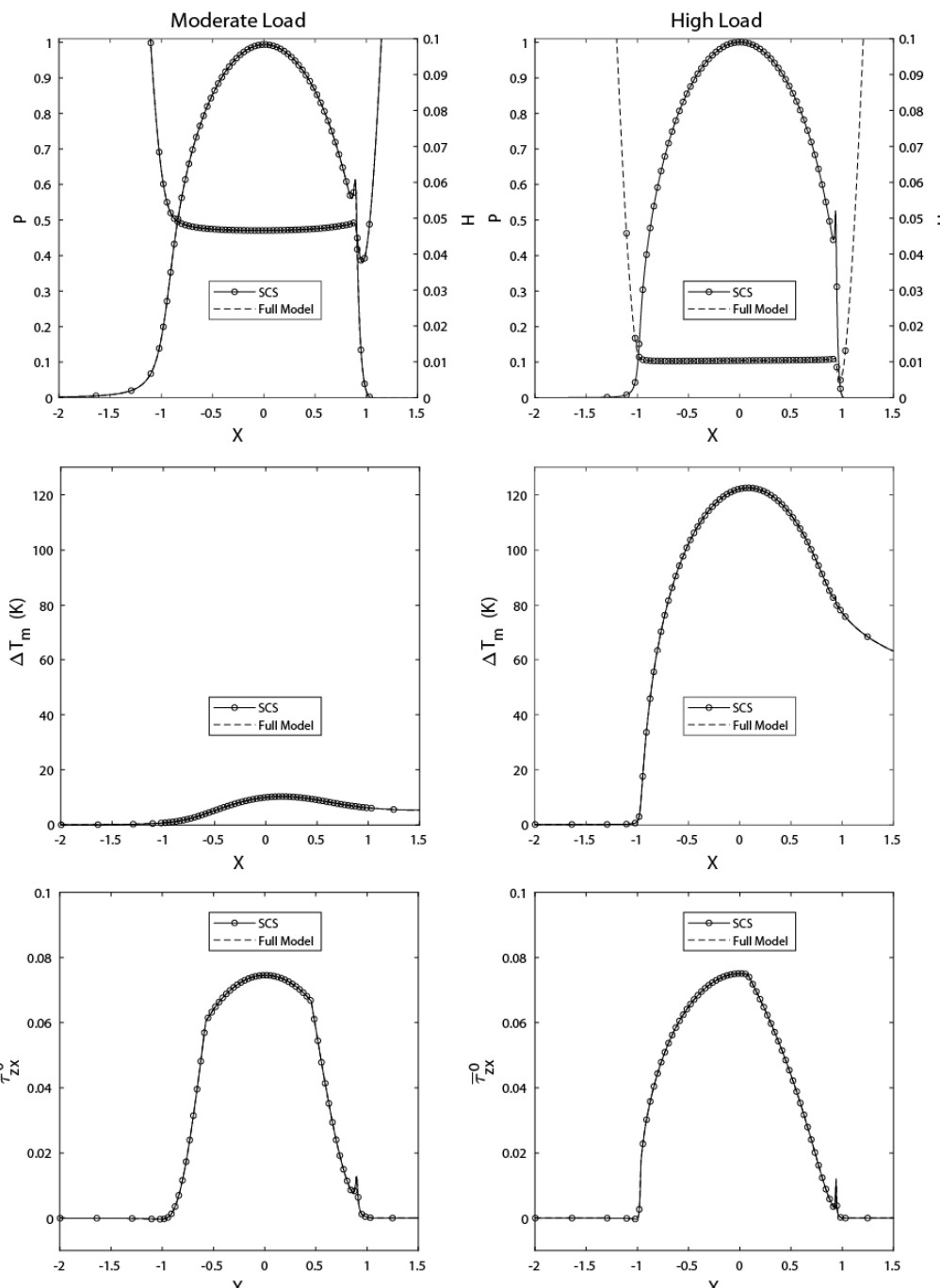

**Figure 6.** Dimensionless pressure and film thickness distributions over the contact domain (top), temperature rise within the mid-layer of the lubricant film (middle), and dimensionless shear stress variations over the plane surface (Z = 0) for the line contact case with *SRR* = 0.5.

The friction curves for both the moderate and high load cases, obtained using both the full and reduced models, can be seen in Figure 7. Note, again, the perfect agreement between the friction curves obtained using both models. The main difference is in the cpu time required for each. For example, for the friction curves of Figure 7, each of which contains 32 data points, the required cpu time differences are reported in Table 2.

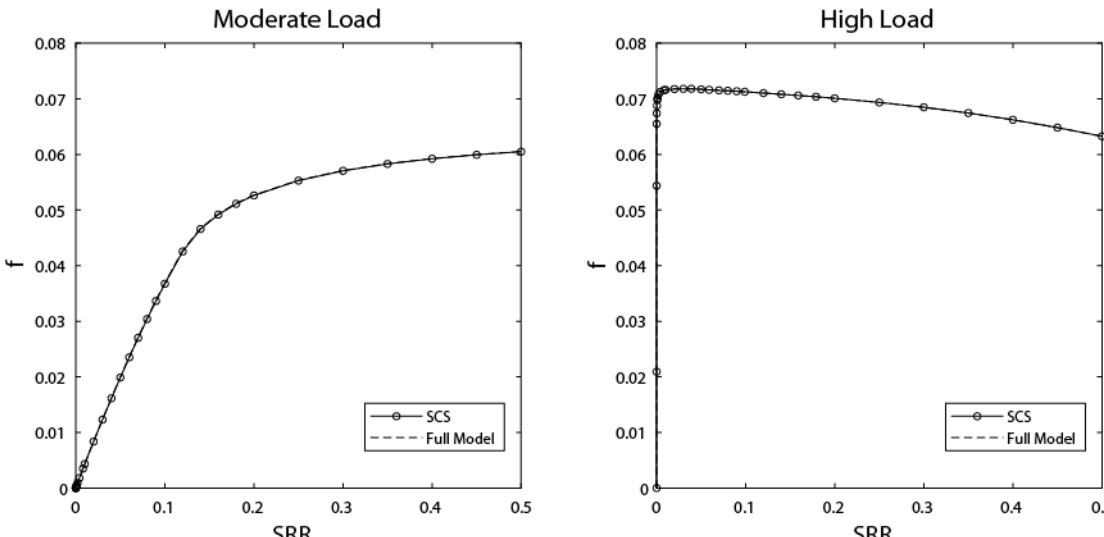

**Figure 7.** Line contact friction curves for moderate and high loads.

**Table 2.** Computational overhead comparison between full and reduced models for generating the line contact friction curves of Figure 7 (32 data points each).

| Load | Full Model | | SCS Model | |
|---|---|---|---|---|
| | # Newt. Iter. | cpu Time (s) | # Newt. Iter. | cpu Time (s) |
| Moderate | 90 | 126.6 | 119 | 118.4 |
| High | 421 | 487.3 | 436 | 398.8 |

It is clear that, despite an increase in the overall number of Newton iterations, cpu times for the reduced model are reduced by 10–20%, compared to the full model.

### 5.2. Point Contacts

The results were computed using the "Normal" mesh defined in [20]. This mesh corresponds to 18,020 tetrahedral elements used to discretize the 3D domain $\Omega$. The projection of these elements on the contact region was used in discretizing the 2D contact domain (to avoid mapping), resulting in a total of 5390 triangular elements for $\Omega_c$. On the other hand, 16,462 tetrahedral elements were used to discretize the plane domain $\Omega_1$. The same number was used for the ball domain $\Omega_2$, of which the mesh is nothing else but a mirror image of that of the plane with respect to the mid-layer of the lubricant film ($Z = 0.5$). Lastly, prism elements were used to discretize the fluid domain $\Omega_f$ of the thermal part, for which projection over the contact domain $\Omega_c$ is nothing but its corresponding 2D triangular mesh. For the normal mesh case used here, the total number of elements across the film thickness is 4. Accordingly, the mesh used contains a total of 10,909 nodes for the hydrodynamic and shear stress parts (10,909 and 21,818 dofs, respectively), 30,533 nodes (91,599 dofs) for the elastic part, and 132,693 nodes/dofs for the thermal part, adding up to a total of $n_{\text{dof}} = 257,020$. Accordingly, the reduced number of degrees of freedom after static condensation is $\hat{n}_{\text{dof}} = 176,330$. Figure 8 shows along the central line of the contact in the $x$-direction, the dimensionless pressure and film thickness distributions over the contact domain (top), the temperature rise within the mid-layer of the lubricant film (middle), and variations of the $x$-component of the dimensionless shear stress over the plane surface ($Z = 0$), for both considered loading conditions (moderate and high) with $SRR = 0.5$, obtained using both the full and reduced models. Clearly, perfect agreement is obtained between the results of the two models. Note that, similar to line contacts, compared to the isothermal Newtonian case (not shown here), the film thicknesses reported in Figure 8 under thermal non-Newtonian considerations are lower due to the combined effects of inlet

shear-thinning, as well as compressive and shear heating. These also reduce the lubricant shear stress across the film, compared to the isothermal Newtonian case. The effect on pressure would be minimal though, except in the vicinity of the pressure spike, which is usually thinner and higher in the isothermal Newtonian case.

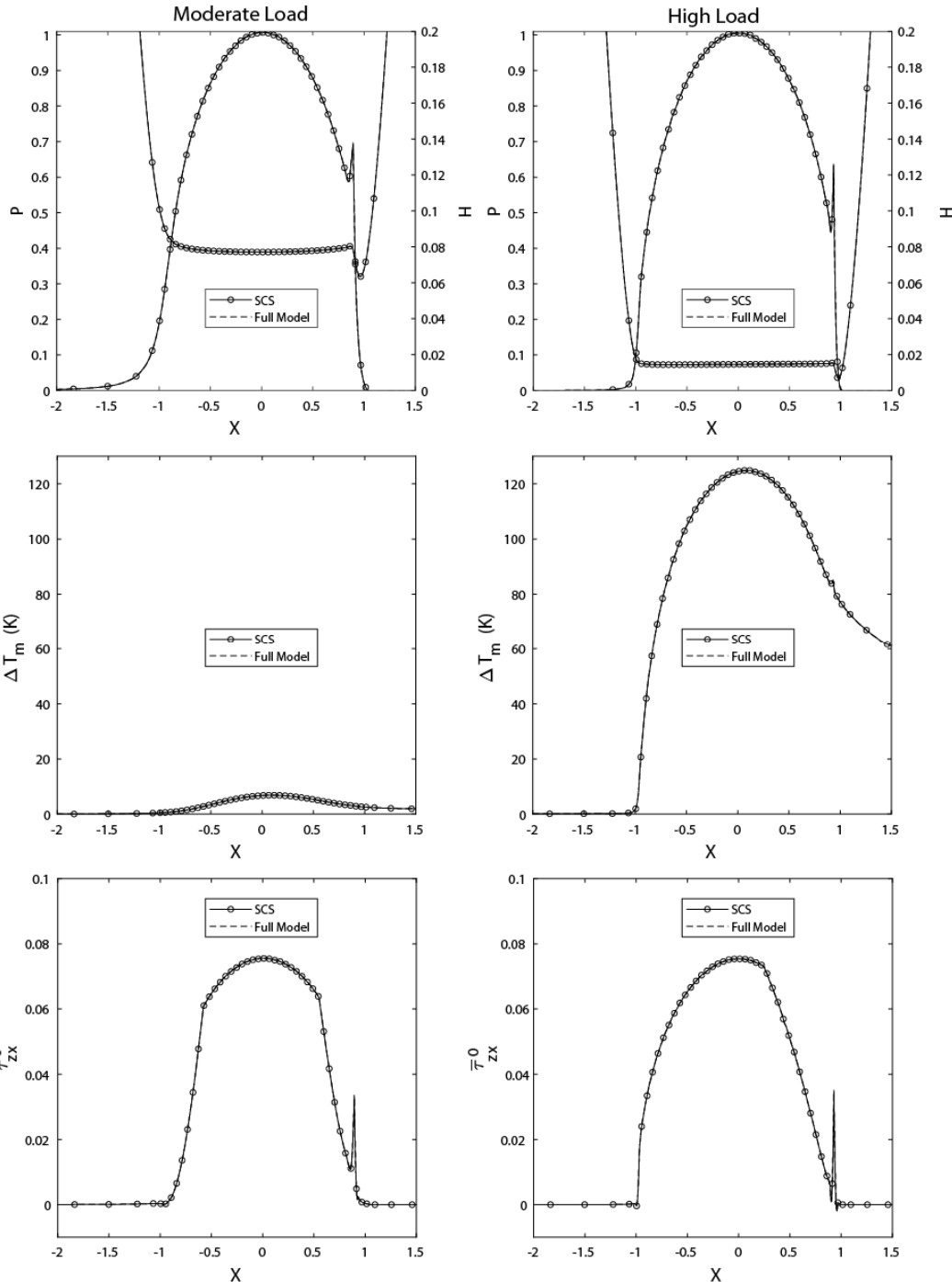

**Figure 8.** Dimensionless pressure and film thickness distributions over the contact domain (top), temperature rise within the mid-layer of the lubricant film (middle), and variations of the *x*-component of the dimensionless shear stress over the plane surface (Z = 0) for the circular contact case with *SRR* = 0.5, along the central line of the contact in the *x*-direction.

The friction curves obtained using the full and reduced model for the moderate and high load cases are also reported in Figure 9, where for circular contacts, the friction coefficient is defined as:

$$f = \frac{2 \times \int\limits_{\Omega_c} \tau_{zx}|_{z=h/2} d\Omega}{F} \tag{41}$$

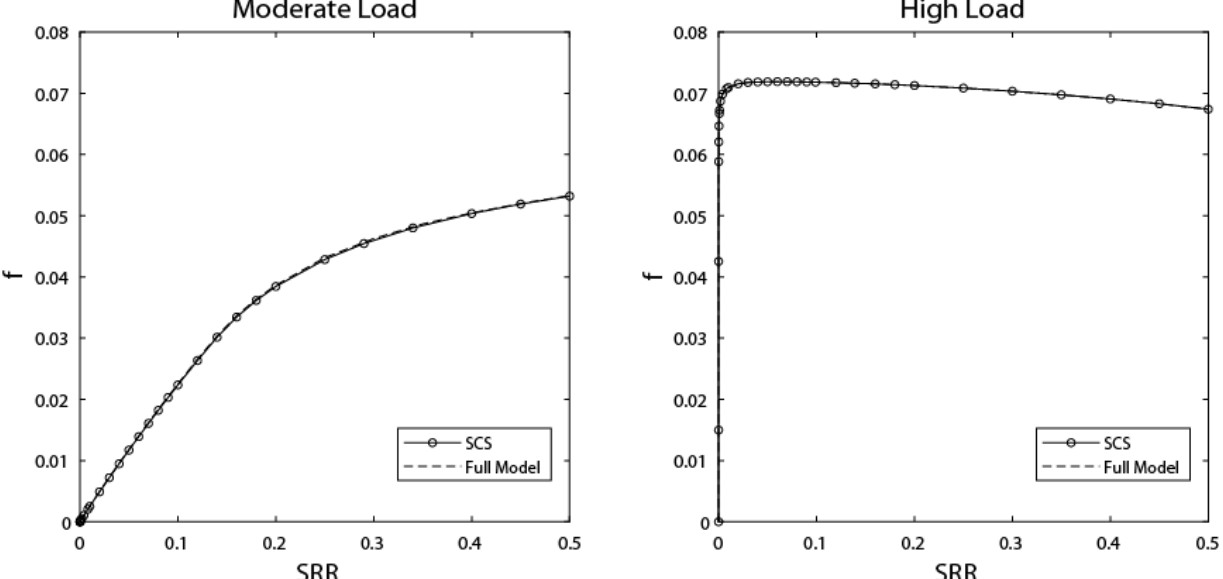

**Figure 9.** Circular contact friction curves for moderate and high loads.

The exactness of the method can be observed again through the friction curves of Figure 9. On the other hand, the main difference is in the cpu time required by each model. For the friction curves of Figure 9 (containing 32 data points each), the reduction in cpu times can be observed in Table 3.

**Table 3.** Computational overhead comparison between full and reduced models for generating the circular contact friction curves of Figure 9 (32 data points each).

| Load | Full Model | | SCS Model | |
|---|---|---|---|---|
| | # Newt. Iter. | cpu Time (s) | # Newt. Iter. | cpu Time (s) |
| Moderate | 85 | 107,356 | 102 | 49,937 |
| High | 451 | 572,022 | 459 | 257,302 |

It is clear that, despite an increase in the overall number of Newton iterations, cpu times for the reduced model are reduced by more than 50%, compared to the full model.

## 6. Conclusions

The derivation of fast, reliable, and accurate modeling procedures for the solution of thermal elastohydrodynamic lubrication problems is a topic of significant interest to the Tribology community. This paper presents a novel model order reduction technique for the finite element modeling of thermal elastohydrodynamically lubricated contacts. The method consists of a static condensation procedure, followed by a splitting algorithm. Static condensation is first used to reduce the size of the linear elasticity part within the overall matrix system by limiting the linear elasticity dofs to the required ones (i.e., normal deformations within the contact domain). However, the ensuing matrix system is semi-dense, requiring a large computational overhead to be inverted. Accordingly, a splitting algorithm is used to retrieve a standard finite-element-like sparsity pattern,

making this method significantly faster than the full model. Even though the reduction in computational times is not as attractive as for traditional MOR techniques, the proposed methodology offers several advantages. Firstly, the offline phase of this method is quite simple since it only requires basic knowledge of linear algebra. In addition, the method is exact since it does not require any approximations or mode superposition as in traditional MOR techniques. As such, this method is versatile, allowing the incorporation of new features (just as thermal and non-Newtonian effects were introduced in this work), without the need to redefine the reduced solution space. In addition, it can be applied to any TEHL model where the solid deformation part is based on the classical linear elasticity equations (e.g., the current approach, or ones that are CFD-based, with a Fluid–Structure Interaction approach [19]). Application of the simple linear algebra operations described here allows for restricting the linear elasticity part to the solid surface. Sub-surface degrees of freedom would be removed, and their effect would be injected into the surface ones, allowing significant reductions in memory usage and computational times. Compared to traditional TEHL modeling techniques, the current approach offers the advantage of fast convergence rates, requiring a few iterations only for a given test case, due to full coupling (i.e., the simultaneous resolution of all TEHL governing equations). In addition, the applied MOR technique means that the size of the arising linear matrix system at every iteration is reduced, requiring a lower computational overhead for its inversion, as well as lower memory requirements. Lastly, the proposed methodology is independent of the material properties of the solids and the operating conditions, which means that the reduction needs to be performed only once for a given mesh case, and the result may be stored for later use.

The performance of the static condensation with the splitting method was tested in this work for both moderate and high load configurations. The obtained results proved the robustness of the method, showing its ability to attain solutions for relatively high loads (known to be numerically challenging). The results also revealed the exactness of the method, which does not introduce any additional model-reduction approximations to the overall solution, since the reduced model results exactly matched those of the full model. The latter has been validated on numerous occasions against experiments (see references [48–51], for example), which validates the current reduced model. On the other hand, the reduction in computational times with respect to the full model was shown to be significant. It is in the order of 10–20% for line contacts, while it is in excess of 50% for circular contacts.

**Author Contributions:** Conceptualization, W.H.; methodology, W.H.; software, J.M.; validation, J.M.; formal analysis, J.M.; investigation, J.M.; resources, W.H.; data curation, J.M.; writing—original draft preparation, J.M.; writing—review and editing, W.H.; visualization, W.H.; supervision, W.H.; project administration, W.H.; funding acquisition, none. All authors have read and agreed to the published version of the manuscript.

**Funding:** This research received no external funding.

**Data Availability Statement:** The data presented in this study are available in article.

**Conflicts of Interest:** The authors declare no conflict of interest.

## Nomenclature

| | |
|---|---|
| $\alpha$ | Lubricant viscosity-pressure coefficient ($Pa^{-1}$) |
| $\beta$ | Lubricant viscosity-temperature coefficient ($K^{-1}$) |
| $\eta$ | Lubricant generalized-Newtonian viscosity ($Pa \cdot s$) |
| $\overline{\eta}$ | Lubricant dimensionless generalized-Newtonian viscosity |
| $\overline{\eta}_e$ | Dimensionless first-order cross-film lubricant viscosity integral |
| $\overline{\eta}'_e$ | Dimensionless second-order cross-film lubricant viscosity integral |
| $\eta_R$ | Lubricant viscosity at reference state ($Pa \cdot s$) |
| $\gamma$ | Dowson and Higginson EoS density-temperature coefficient ($K^{-1}$) |

| | |
|---|---|
| $\dot{\gamma}_{ij}$ | Lubricant shear rate in the $j$-direction within a plane having $i$ as normal ($\text{s}^{-1}$) |
| $\mu_R$ | Lubricant low-shear/Newtonian viscosity at reference state (Pa·s) |
| $\overline{\mu}_1, \overline{\mu}_2$ | Dimensionless lubricant first and second Newtonian viscosities |
| $\mu_{1,R}, \mu_{2,R}$ | Lubricant first and second Newtonian viscosities at reference state (Pa·s) |
| $v$ | Equivalent solid Poisson coefficient |
| $v_1, v_2$ | Poisson coefficient of solids 1 and 2 |
| $\Omega$ | Equivalent solid computational domain |
| $\Omega_c$ | Contact computational domain |
| $\Omega_f$ | Lubricant film computational domain within thermal part |
| $\Omega_1, \Omega_2$ | Computational domain of solids 1 and 2 within thermal part |
| $\partial\Omega_{cs}$ | Symmetry boundary of $\Omega_c$ |
| $\partial\Omega_b$ | Fixed boundary of $\Omega$ |
| $\partial\Omega_s$ | Symmetry boundary of $\Omega$ |
| $\Lambda$ | Lubricant limiting shear stress-pressure coefficient |
| $\rho$ | Lubricant density ($\text{kg/m}^3$) |
| $\rho_1, \rho_2$ | Density of solids 1 and 2 ($\text{kg/m}^3$) |
| $\overline{\rho}$ | Lubricant dimensionless density |
| $\overline{\rho}_e$ | Dimensionless first-order cross-film lubricant density integral |
| $\overline{\rho}'_e$ | Dimensionless first-order cross-film density-to-viscosity double-integral |
| $\overline{\rho}''_e$ | Dimensionless second-order cross-film density-to-viscosity double-integral |
| $\rho_R$ | Lubricant density at reference state ($\text{kg/m}^3$) |
| $\sigma_n$ | Normal component of 2D or 3D stress tensor (Pa) |
| $\sigma_t$ | Tangential component of 2D stress tensor (Pa) |
| $\{\sigma_t\}$ | Vector of tangential components of 3D stress tensor (Pa) |
| $\tau$ | Lubricant resultant shear stress (Pa) |
| $\overline{\tau}^0$ | Lubricant dimensionless resultant shear stress over plane surface |
| $\tau_L$ | Lubricant limiting shear stress (Pa) |
| $\tau_R$ | Reference shear stress (Pa) |
| $\tau_{ij}$ | Shear stress in the $j$-direction within a plane having $i$ as normal (Pa) |
| $\overline{\tau}_{ij}$ | Dimensionless shear stress in the $j$-direction within a plane having $i$ as normal |
| $\overline{\tau}^0_{ij}$ | Dimensionless lubricant shear stress $\overline{\tau}_{ij}$ over plane surface |
| $\theta$ | Heaviside function |
| $\xi$ | Penalty term parameter |
| $a$ | Hertzian contact half-width (line contact) or radius (circular contact) (m) |
| $a_c, n_c$ | Double-Newtonian modified Carreau model parameters |
| $c_1, c_2$ | Specific heat of solids 1 and 2 (J/kg·K) |
| $E$ | Equivalent solid Young's modulus of elasticity (Pa) |
| $E_1, E_2$ | Young's moduli of elasticity of solids 1 and 2 (Pa) |
| $f$ | Friction coefficient |
| $F$ | Contact external applied load (N/m: line contacts or N: point contacts) |
| $G_c$ | Lubricant critical shear stress (Pa) |
| $h$ | Lubricant film thickness (m) |
| $H_0$ | Dimensionless rigid-body separation |
| $H$ | Dimensionless lubricant film thickness |
| $k$ | Lubricant thermal conductivity (W/m·K) |
| $k_1, k_2$ | Thermal conductivities of solids 1 and 2 (W/m·K) |
| $n_{dof}$ | Total number of degrees of freedom of FEM model |
| $\hat{n}_{dof}$ | Total number of degrees of freedom of reduced FEM model |
| $n_m, n_s$ | Numbers of master and slave dofs in reduced FEM model |
| $p$ | Pressure (Pa) |
| $p_h$ | Hertzian contact pressure (Pa) |
| $p_R$ | Reference pressure (Pa) |
| $P$ | Dimensionless pressure |
| $Q_{comp}$ | Compressive heat generation per unit volume ($\text{W/m}^3$) |
| $Q_{shear}$ | Shear heat generation per unit volume ($\text{W/m}^3$) |
| $R$ | Equivalent roller radius (m) |

| $SRR$ | Slide-to-roll ratio |
|---|---|
| $T$ | Temperature (K) |
| $\overline{T}$ | Dimensionless temperature |
| $T_R$ | Reference temperature (K) |
| $T_0$ | Ambient temperature (K) |
| $u_1, u_2$ | Surface $x$-velocity components of solids 1 and 2 (m/s) |
| $u_f, v_f$ | Lubricant velocity field components in the $x$, $y$-directions (m/s) |
| $u, v, w$ | Equivalent solid deformation components in the $x$, $y$, $z$-directions (m) |
| $u_m$ | Contact mean entrainment speed in the $x$-direction (m/s) |
| $U, V, W$ | Equivalent solid dimensionless deformation components in $x$, $y$, $z$-directions |
| $\overline{U}$ | Equivalent solid dimensionless deformation vector |
| $x, y, z$ | Space coordinates (m) |
| $X, Y, Z$ | Dimensionless space coordinates |
| **Subscripts** | |
| 1 | Flat plane |
| 2 | Cylinder (line contacts)/ball (point contacts) |
| $f$ | Fluid domain |
| $e$ | Elastic domain |
| $h$ | Hydrodynamic domain |
| $l$ | Load balance domain |
| $m$ | Master dofs |
| $s$ | Shear stress domain/Slave dofs |
| $t$ | Thermal domain |
| **Superscripts** | |
| 0 | Plane surface |
| $f$ | Far dofs |
| $n$ | Near dofs |

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
