# Peer review of "Exact Model Order Reduction for the Full-System Finite Element Solution of Thermal Elastohydrodynamic Lubrication Problems"

_lubricants, doi:10.3390/lubricants11020061_

Round 1

Reviewer 1 Report

This paper is well written and introduce a novel model order reduction technique for the analysis of thermal elastohydrodynamic lubrication problems. The topic is interesting and valuable.

Minor comment

1.        Nomenclature should be provided since there are too many symbols.

2.        In Figs.6,7and8,simulation results of SCS and full model agree much well, it is better to make it more clearly. For example, square, triangle and other symbols

Author Response

The authors wish to thank the reviewer for his/her valuable comments and suggestions. These have been taken into consideration in the revised manuscript and the answers to the different questions and comments raised by the reviewer are provided below (in red). Amendments to the original text as well as any additions are highlighted in yellow to facilitate the track of changes process in the revised version.

Reviewer 2 Report

In principle, I have no substantive comments on the content of the article. The presented method using the split algorithm is much faster than the full model and provides many advantages. The question is how to easily adapt this method to existing numerical systems and compare it with them as to pratctical effects.  

Author Response

(The authors gave the same response as above.)

Reviewer 3 Report

There are some significant contributions to TEHL that the author has not cited those. Generally, the introduction needs improvement by including some of the original contributions to TEHL. A simple search in good scholar will show the author the relevant publications, particularly those that have applied the TEHL to real applications.

I am surprised to see that the author uses Roelands' formula. As is shown by others, Roeland's formula significantly underpredicts viscosity particularly at moderate and particularly high contact pressures which is subject of the current paper.

A nomenclature section must be included.

There is no reliable independent validation of the results. The results are validated with the results from previous numerical model of the authors but there is a requirement for validating the results against experimental results.

Please comment on the inlet shear heating effect and its inclusion in the current analysis.

Author Response

(The authors gave the same response as above.)

Round 2

Reviewer 3 Report

Thank you for addressing my comments.